# Towards a Unified Approach in Autoimmune Fibrotic Signalling Pathways

**DOI:** 10.3390/ijms24109060

**Published:** 2023-05-21

**Authors:** Margherita Sisto, Sabrina Lisi

**Affiliations:** Department of Translational Biomedicine and Neuroscience (DiBraiN), Section of Human Anatomy and Histology, University of Bari “Aldo Moro”, Piazza Giulio Cesare 1, I-70124 Bari, Italy; sabrina.lisi@uniba.it

**Keywords:** autoimmunity, inflammation, fibrosis

## Abstract

Autoimmunity is a chronic process resulting in inflammation, tissue damage, and subsequent tissue remodelling and organ fibrosis. In contrast to acute inflammatory reactions, pathogenic fibrosis typically results from the chronic inflammatory reactions characterizing autoimmune diseases. Despite having obvious aetiological and clinical outcome distinctions, most chronic autoimmune fibrotic disorders have in common a persistent and sustained production of growth factors, proteolytic enzymes, angiogenic factors, and fibrogenic cytokines, which together stimulate the deposition of connective tissue elements or epithelial to mesenchymal transformation (EMT) that progressively remodels and destroys normal tissue architecture leading to organ failure. Despite its enormous impact on human health, there are currently no approved treatments that directly target the molecular mechanisms of fibrosis. The primary goal of this review is to discuss the most recent identified mechanisms of chronic autoimmune diseases characterized by a fibrotic evolution with the aim to identify possible common and unique mechanisms of fibrogenesis that might be exploited in the development of effective antifibrotic therapies.

## 1. Introduction

The pathophysiology of autoimmune diseases involves an unbalanced interplay between the innate and adaptive immunity, culminating in immune responses mounted against self-antigens [1]. All autoimmune diseases are believed to go through sequential initiation, effector, and resolution phases. In contrast to normal inflammation, in autoimmune diseases there is sustained cellular activation, resulting in chronic inflammation. Recently, increasing evidence show that the abnormal inflammatory response is closely associated with many chronic autoimmune diseases, including rheumatoid arthritis (RA), systemic lupus erythematosus (SLE), systemic sclerosis, Sjӧgren’s syndrome, and diabetes, with concomitant tissue damage, remodelling, and, as end result, organ fibrosis [2,3]. Fibrosis is a pathological feature of most chronic inflammatory autoimmune diseases. The underlying molecular and cellular events of fibrotic autoimmune diseases share many functional similarities, despite differences in aetiology and clinical evolution. Although fibrosis is initially beneficial, the repair process can become pathogenic, and the continuous accumulation of fibrotic proteins leads to permanent tissue remodelling and significant organ dysfunction and failure. There are multiple versions of fibrotic autoimmune disorders that affecting a variety of organs, and it has been estimated that a significant percentage of deaths can be attributed to organ fibrotic transformation [4]. The molecular events and processes which contribute to the onset and development of autoimmune-associated fibrosis must be elucidated in order to develop novel efficacious treatments. This review was born with the intention of collecting the most recent contributions that can clarify the cellular and molecular mechanisms at the basis of the fibrotic evolution of various autoimmune diseases. The review focuses on the intertwined pathophysiological role of fibrosis and chronic inflammation in different tissues and autoimmune diseases, with the aim of identifying possible molecular bridges between the various fibrotic mechanisms identified in autoimmune diseases.

## 2. Molecular Mechanisms Involved in the Fibrotic Evolution during Autoimmune Diseases

Within the recent research topics, several experimental works on the cellular and molecular mediators of fibrosis in autoimmune diseases were produced, and the intertwined pathophysiological roles of fibrosis and chronic inflammation were demonstrated. The following paragraphs illustrate the mechanisms identified thus far for the fibrotic evolution during autoimmune diseases, in order to identify mediators common to several mechanisms and build connecting bridges between them.

### 2.1. Growth Factors and Associated Signalling Pathways

Inputs from growth factors converge on several signalling pathways to promote fibrosis inducing fibroblast activation, epithelial cell apoptosis, epithelial to mesenchymal transformation (EMT), and endothelial to mesenchymal transition (EndMT). The growth factors involved in these fibrotic pathways are represented by TGF-β factors, platelet-derived growth factors (PDGFs), fibroblast growth factors (FGFs), vascular endothelial growth factor (VEGF), and connective tissue growth factor (CTGF). Subsequent downstream signalling pathways are activated such as phosphatidylinositol 3-kinase (PI3K)/protein kinase B (AKT), Janus kinase (JAK)/signal transducer and activator of transcription (STAT), ADAM17 metalloproteinases, and WNT/β-catenin. The following paragraphs will illustrate the fibrotic mechanisms dependent on the activation of growth factors involved in the pathogenesis of various autoimmune diseases. The recently identified signalling pathways involved in fibrosis are depicted in Figure 1.

#### 2.1.1. TGF-β Signalling Pathway

TGF-βs are the key cytokines in most fibrosis pathways. TGF-β is secreted from cells in a latent tripartite complex consisting of its dimeric inactive form (named latency-associated peptide [LAP]) plus a latent TGF-β-binding protein (LTBP) [5]. This TGF-β/LAP/LTBP inactive complex binds to the ECM components [5]. After cleavage by various proteases, the active form of TGF-β is released [5] and is able to bind to TGFβR2 and TGFβR1 receptors [6]. TGF-β expression is promoted by epidermal growth factor (EGF), IL-1, and TNF-α [7]; moreover, integrin αv/β6 could determine the activation of the TGF-β gene, due to the presence of an arginine–glycine–aspartate (RGD) motif in the precursors of TGF-β [8]. Experimental results demonstrated that by blocking integrin αv/β6 using a specific antibody, pulmonary fibrosis in mice was prevented and the inflammatory situation did not undergo evolution [9]. Two signalling pathways, known as Smad-mediated canonical and non-canonical, are activated by TGF-βs to regulate fibrotic evolution [7,10]. In the canonical pathway, after TGFβR1 activation, the Smad2 and Smad3 intracellular effector are phosphorylated; the subsequent interaction with Smad4 determines the nuclear transcription of target genes [11]. TGF-β can also activate Smad-independent non-canonical signalling pathways, through the activation of PI3K/AKT, mitogen-activated protein kinases (MAPKs), and JAK/STAT proteins [12]. Although fibroblasts are major sources and targets of TGF-β, some fibrogenic transformations reflect the activation of other cell types such as macrophages and epithelial cells [13]; in addition, the promotion of TGF-β-dependent fibrosis in autoimmune diseases involves the activation of diverse mechanisms that will be clarified in the next paragraphs.

##### TGF-β-Mediated Fibrosis through Activation of Resident Fibroblasts

The canonical TGF-β1/Smad3 signalling pathway initially determines the recruitment of inflammatory cells and fibroblasts into sites of injury and promotes the differentiation of fibroblasts to myofibroblasts which display exaggerated ECM production [14]. Several authors demonstrated that reactive oxygen species (ROS) mediate TGF-β-induced activation and transformation of fibroblasts. This involves the activation of NADPH oxidase (Nox) enzymes responsible for ROS formation [15]. Nox expression is, in turn, mediated by TGF-β. Nox4 expression, for example, can be induced by TGF-β in a variety of cells [15]. In primary human cardiac fibroblasts, TGF-β1 treatment increased the level of Nox4 and alpha-smooth muscle actin (α-SMA), a myofibroblast marker, whereas depletion of Nox4 decreased TGF-β1-stimulated α-SMA expression, demonstrating how the expression and activation of Nox and TGF-β are dependent on each other [16]. Although there is no certain evidence of the involvement of this ROS-mediated TGF-β1-induced fibrosis in autoimmune myocarditis, numerous preliminary studies seem to point towards this discovery.

##### TGF-β Determines Fibrosis through the Induction of Apoptosis

TGF-β1-dependent fibrosis and apoptosis are juxtaposed although the mechanisms might differ between different cell types. Endothelial cell apoptosis is mediated by ROS production which is dependent on TGF-β activation and mediated by p38 [17]. TGF-β1 could also induce apoptosis of mesangial cells in the kidney via p53 phosphorylation and up-regulation of the pro-apoptotic protein Bcl-2-associated protein X (Bax) [18]. In addition, there is evidence that fibroblasts from fibrotic tissues are resistant to apoptosis, and that TGF-β may confer resistance to apoptosis by classic death pathways such as the Fas–caspase activation cascade [19,20]. This mechanism may be mediated through the activation of signalling pathways, such as p38 MAPK and PI3K–Akt [21], and involve regulation of “inhibitor of apoptosis” (IAP) family members [22,23]. TGF-β may also promote fibroblast resistance to apoptosis through cell-cycle regulators such as p14ARF [24]. Inhibition of fibroblast and myofibroblast apoptosis through these pathways may explain the accumulation of these cells during fibrosis. Myofibroblasts are integral in a feedback loop that perpetuates fibrosis through the stiffening of the extracellular matrix. Lagares et al. determined that stiffness-activated myofibroblasts show an increased expression of pro-apoptotic proteins, and these cells become dependent on antiapoptotic protein expression to prevent their death [25]. A drug that mimics a pro-apoptotic protein blocking the anti-apoptotic protein BCL-XL was demonstrated to induce apoptosis in fibroblasts from patients with scleroderma, (also known as systemic sclerosis), a systemic autoimmune disease that often leads to fibrosis in the skin, heart, vasculature, and lungs [26]. In fact, the inhibition of anti-apoptotic BCL-2 proteins with a specific drug (ABT-263) resulted in effective reversal of fibrosis in a mouse model of scleroderma [27]. Based on these results, the block of anti-apoptotic proteins to induce myofibroblast apoptosis could be an effective strategy to treat fibrosis. A schematic representation of TGF-β mechanisms leading to the activation of fibrosis is shown in Figure 2.

##### TGF-β Regulates Fibrosis through the Activation of EMT

The most common type of EMT program activated in fibrotic evolution is type 2 EMT. Type 2 EMT, mainly triggered by chronic inflammation, is closely linked to the tissue damage repair response and gives rise to myofibroblasts through epithelial cell transition [28]. TGF-β is the main actor in this process, encouraging fibroblast proliferation and migration, promoting fibroblast phenotypic modifications into myofibroblasts and regulating type 2 EMT-dependent fibrosis [7]. TGF-β-mediated EMT-dependent fibrosis is regulated by various factors. Oxidative stress induced by TGF-β is a key event in the EMT process. ROS initiate several effects of TGF-β, influencing several downstream TGF-β signal transduction mediators, including Smads, MAPKs, and NF-κB [7]; TGF-β, on the other hand, increases the level of ROS by upregulating the expression of Nox4 and activating ERK and mTOR signalling-dependent EMT and fibrosis [29]. PI3K/AKT signals also mediate TGF-β-induced EMT [30]. PI3K/Akt and MAPK are two Smad-independent pathways induced by TGF-β1 that contribute to EMT-dependent fibrosis [30]. With this large and complicated scenario in autoimmunity, it is currently widely accepted that TGF-β1-mediated EMT-dependent fibrosis represents the evolution of various chronic inflammatory autoimmune diseases. Various researchers have demonstrated an altered expression of pro-fibrotic molecules in the joints of patients affected by RA [31]. In RA, TGF-β1 acts both as a pro-angiogenic molecule, predominantly in the synovial membrane [32], and as an activator of the synthesis of pro-inflammatory cytokines [32], metalloproteinases [33], and various fibrinolytic factors responsible for tissue remodelling, such as aggrecanase [34] and urokinase-type plasminogen activator [35]. In addition, in RA patients, the loss of inhibitory Smad7 was associated with marked activation of TGFβ/Smad3-dependent EMT [36]. Confirming the inhibitory role of Smad7 in tissue fibrosis and inflammation, the Smad7 deficiency observed in RA patients led to enhanced NF-κB activity, Th1/Th17 cell differentiation, and exacerbation of synovial inflammation, probably through the hyperactivation of the TGF-β/Smad3–IL-6 molecular pathway [36]. In RA, the link between TGF-β expression and EMT was recently confirmed through the observation that the synovial membrane or synovial fluid showed increased levels of TGF-β [37]; furthermore, EMT features, such as more aggressive and invasive cell phenotypes and resistance to apoptosis, that causes pannus tissue invasion and destruction in RA were detected [38]. Some highly innovative studies reported that, in RA, TGF-β is up-regulated by transglutaminase 2 (TG2), an enzyme that regulates ECM composition and degradation and, consequently, EMT through the activation of the transcription factors Snail/Slug, Twist, Zeb, and E47 [39]. In recent years, the activation of an EMT program related to fibrotic evolution has been demonstrated for various other autoimmune diseases, such as inflammatory bowel disease (IBD) [40], ulcerative colitis [41], and Crohn’s disease [42]. Additionally, renal fibrosis features are often linked to SLE nephritis [43]. Similarly, in autoimmune diabetes, complications that involve the lung are present, characterized by the induction and progression of fibrosis into the pulmonary tissue [44]. Diabetes, through the persistent effects of hyperglycaemia, can induce EMT-dependent fibrosis that is the result of inflammatory cell infiltrations into the lung and of the expression of elevated levels of ECM proteins, which induce an inflammatory process leading, consequently, to fibrosis and tissue injury [44]. This process is modulated mainly by the overexpression of TGF-β1 and the EMT transcriptional factor Snail; the elevated accumulation of ECM proteins in the pulmonary tissue is accompanied by the downregulation of epithelial markers such as ZO-1 and cadherin [45]. In autoimmune diabetes, EMT-induced fibrosis occurs via the activation of both SMAD-dependent and SMAD-independent pathways, as demonstrated by the increased levels of expression of the TGF-β1 receptor and SMAD2/3 protein in the diabetic cells, as well as increased levels of p38 and ERK [44]. Additionally, in multiple sclerosis, Troletti and colleagues hypothesized that TGF-β1-induced brain endothelial cell dysfunction might be due to brain endothelial cell trans-differentiation through EndoMT [45]. In multiple sclerosis lesions, TGF-β1 significantly boosted Snail mRNA and protein levels as well as the levels of mesenchymal markers such as fibronectin and vimentin [45,46]. On the contrary, the mRNA levels of junctional factors, such as claudin-1 and claudin-5, were significantly reduced. The triggering of EMT-dependent fibrosis in a situation of chronic inflammation, was recently demonstrated in the chronic inflammatory autoimmune disease Sjögren’s syndrome that affects salivary glands [10,47]. Salivary gland biopsies of patients affected by Sjögren’s syndrome show increased expression of TGF-β1. TGF-β1 stimulates salivary gland epithelial cells leading to phosphorylation and activation of Smad2/3, which form heterocomplexes with Smad4, triggering the canonical Smad-mediated EMT cascade. Interestingly, blocking canonical TGF-β1/Smad2/3 signal transduction had no effect on the activation of the non-canonical TGF-β1/Erk1/2/EMT pathway, suggesting that, in Sjögren’s syndrome, both the canonical and non-canonical signalling pathways are activated independently to induce EMT-dependent fibrosis [10,47]. Figure 3 shows the TGF-β-mediated mechanisms involved in EMT-dependent fibrotic evolution of autoimmune diseases.

#### 2.1.2. PDGFs/PDGFRs

PDGFs are stimulators of cell division that are required for cell growth and proliferation of specific tissues. PDGF ligands bind to their receptors PDGFRαα, PDGFRαβ and PDGFRββ [48] activating downstream signals (RAS/MAPK, PI3K/AKT, and JAK/STAT pathways) [49]. PDGFs are mainly produced by macrophages, endothelial cells, and fibroblasts [50]. PDGFs and PDGF receptors are increased in the fibrotic skin lesions observed in autoimmune systemic diseases [51,52]. Furthermore, both PDGF-B and PDGF-D act as potent factors for hepatic stellate cell proliferation and migration, determining ECM deposition during liver fibrogenesis [53]. However, interestingly, PDGF-C seems to not be involved in liver fibrosis or functional liver impairment, or, as reported by some authors, PDGF-C and PDGF-D seem to be involved in the late stage of hepatic fibrogenesis [54]. Several recent studies demonstrated that PDGFs contribute to the formation of heart and lung fibrosis also via stimulating the activation of fibroblasts [55]. The mechanisms linking PDGF to the chronic inflammation observed in autoimmune conditions still remain to be clarified. Indeed, tissue fibrosis accompanying intractable chronic inflammation is associated with increased PDGF signalling and with proliferation and accumulation of PDGFR-positive mesenchymal cells or fibroblasts in the synovium of RA patients, in which a greater expression of phosphorylated active PDGFRαβ receptors was detected [56]. Interestingly, recent papers reported the presence of brain mesenchymal perivascular aggregates of PDGFRβ-positive cells in multiple sclerosis patients, where scar-forming cells can persist chronically in a condition of active inflammation and demyelination which characterize the disease [57].

#### 2.1.3. FGFs/FGFRs

The fibroblast growth factor (FGF) family consists of signalling ligands that bind with variable affinity to four FGF receptors (FGFRs) [58]; FGFs can induce the dimerization, activation, and autophosphorylation of FGFRs and subsequently the activation of the RAS–extracellular signal-regulated kinase (ERK), PI3K–AKT, and JAK/STAT pathways [59]. The role of the FGF family in fibrosis development is not completely clear but was studied in the liver [60,61]. In particular, two isoforms of FGF2 showed very interesting behaviours during hepatic fibrogenesis; the form of FGF2 with a low molecular weight, when used to treat hepatic stellate cells, attenuated fibrosis by down-regulation of Delta-like 1 protein expression through the p38 MAPK pathway. In contrast, the isoform of FGF2 with a high molecular weight promoted hepatic fibrogenesis [62]. In the case of idiopathic pulmonary fibrosis, on the other hand, FGF1/FGFR signalling is aberrantly increased and may promote fibroblast migration via increased MAPK signalling, leading to the pathogenesis of lung fibrosis. Recent studies confirmed that the inhibition of FGF/FGFR signalling can reduce fibrosis in experimental animal models and the inhibition of FGF signalling is important in treating pulmonary fibrosis [63]. In RA, FGF-1 and FGF-2 have been implicated in abnormal synoviocyte proliferation and apoptosis resistance [64]. In addition, the FGF pathway seems to be implicated in interstitial (or diffuse parenchymal) lung diseases (ILDs) that represent a large, heterogeneous group of rare pulmonary pathologies, characterized by damaged parenchyma and mediated by varying degrees of chronic inflammation and fibrosis [65]. ILDs can represent the pulmonary complication of RA [66] and systemic sclerosis [67]. In the case of ILDs, the aetiology is unknown [68].

#### 2.1.4. VEGFs/VEGFRs

The VEGF family comprises six members: VEGF-A, -B, -C, -D, and -E, and placental growth factor (PIGF) [69]. VEGFs regulate vasculogenesis, angiogenesis, and immunological responses [70]. VEGF-A, which exerts its biological functions by binding to VEGFR1 and VEGFR2, is widely studied as an angiogenesis regulator in homeostasis and diseases [71,72]. VEGF-A was decreased in autoimmune idiopathic pulmonary fibrosis patients, and the overexpression of VEGF-A protected lung tissue from damage and fibrosis [73]; Murray et al. [73] have proposed a non-cell autonomous function mediated by the endothelium to explain this epithelial-protective function of VEGF-A. However, there are conflicting reports as to whether VEGF-A is a contributing or protective factor against fibrosis because several clinical studies support the notion that VEGF-A might facilitate pulmonary fibrogenesis, depending on the specific type of VEGF-A isoform expressed in the tissue [74,75]. Systemic sclerosis is a rare autoimmune disease marked by the fibrosis of the skin and involvement of internal organs, especially the vascular system, lungs, kidneys, and gastrointestinal system, and is caused by excessive collagen deposition, immunological disturbances, and accompanying vascular changes. Microvascular damage and dysfunction of angiogenesis are the identified abnormalities in this disease. Tissue fibrosis results from a series of events, including endothelial dysfunction, inflammation, increased vascular permeability, and platelet aggregation [76]. Altered angiogenesis biomarker expression and microvascular damage are detected in the digit ulcers of systemic sclerosis patients [77]. Higher levels of VEGF in comparison to healthy controls were detected in both the early and established stages of systemic sclerosis [78]. Dysregulated tissue remodelling with aberrant fibrosis is one of the pathological hallmarks of autoimmune rheumatic diseases and interstitial lung disease is an important cause of disease-related morbidity across this group of disorders, particularly within connective tissue diseases such as systemic sclerosis [79]. Interstitial lung disease is the leading cause of disease-related mortality in systemic sclerosis [80]. The induction of VEGF pathways by hypoxia [81] has led researchers to consider its potential role in the pathogenesis of systemic sclerosis. Subsequent work examining VEGF-A splice isoforms provided a plausible explanation, having associated increased plasma levels of the VEGF-A165b splice variant with severe nailfold capillary loss [82]. In addition, when interstitial lung disease is associated with systemic sclerosis, lower VEGF-A BALF levels were detected compared to both healthy controls and systemic sclerosis patients without lung involvement [83]. Additionally, in rheumatoid arthritis, circulating VEGF-A is increased in patients’ sera, particularly in those patients with extra-articular manifestations (including pulmonary fibrosis) [84].

#### 2.1.5. CTGF Signalling Pathway

Connective tissue growth factor (CTGF) is a secreted peptide involved in cell proliferation, angiogenesis, and wound healing; it has also been implicated in tumour development and tissue fibrosis [85]. CTGF is considered a factor that enhances PDGF-B signalling, especially in the PDGF-related regulation of the proliferation and chemotaxis of fibroblasts [86,87]. CTGF synthesis is induced by many pro-fibrotic cytokines such as TGFβ and VEGF [88]; CTGF can, in turn, combine with other factors to promote pro-fibrotic effects. TGF-β-mediated endogenous CTGF induction leads to negative regulation of Smad7 gene transcription; since Smad7 is an inhibitor of the canonical Smad-mediated TGF-β pathway, by blocking the inhibitory effect of Smad7, CTGF induces a persistent activation of pro-fibrotic TGF-β signalling [89]. CTGF regulates hepatic stellate cell adhesion, a critical event during fibrogenesis in hepatic fibrosis [90]. In addition, CTGF activates myofibroblast formation by trans-differentiating resident fibroblasts and epithelial cells through the activation of the EMT program [91]. Among various fibrotic diseases, CTGF has been extensively studied in autoimmune idiopathic pulmonary fibrosis [92] and cardiac [93], liver [94], and renal fibrosis [95]. In idiopathic pulmonary fibrosis, the overexpression of CTGF, in cooperation with TGFβ, is profibrotic and exacerbates ECM deposition in mouse lung tissues [96]. The severity of fibrosis was markedly attenuated by CTGF inhibition, confirming the pro-fibrotic activity of CTGF [97]. Although the involvement of CTGF has been well-documented in systemic sclerosis fibrosis, the therapeutic potential in targeting CTGF is still being studied in this autoimmune disease. The data collected evidenced a role for Angiotensin II which seems to induce skin fibrosis that was mitigated after CTGF gene silencing [98]. In CTGF knock out mice, the number of cells expressing PDGFRβ, procollagen, αSMA, pSmad2, CD45, and Fsp1 in the dermis was significantly reduced, suggesting a key role for CTGF in the fibrotic evolution in systemic sclerosis [99].

### 2.2. ADAM17 Activation and Fibrosis in Autoimmune Diseases

ADAM17 is a disintegrin and metalloproteinase (ADAM) family member that acts as a sheddase of various membrane proteins. Over the recent decades, ADAM17 has been reported to be a key factor in several biological pathways regulating proliferation, migration, and the immune response [100]. Therefore, it is not surprising that ADAM17, involved in the pathophysiology of numerous human diseases, is critically implicated in EMT and EMT-dependent fibrosis [101]. Since ADAM17 mediates the ectodomain shedding of various pro-inflammatory molecules, it is of no surprise that ADAM17 has attracted attention as a potential driver of inflammation and is repurposed pathologically during fibrosis [102]. Supporting this concept, high ADAM17 expression was detected in numerous human chronic inflammatory diseases, and it has been hypothesized that ADAM17 represents a convergence point between inflammation and the progression of degenerative EMT-dependent fibrotic diseases [103]. In fact, several recent studies have shown correlations between the increased levels of ADAM17 expression and the severity of fibrosis in patients with degenerative fibrotic diseases [103]. The specific role of ADAM17 in the pathophysiology of chronic inflammatory autoimmune diseases and fibrotic diseases is not fully understood and appears to depend on the cellular context. ADAM17 is a crucial modulator of the pathological airway remodelling in lung diseases, including asthma, chronic obstructive pulmonary disease, and cystic fibrosis [104]. In addition, the overexpression of ADAM-17 was detected in Sjӧgren’s syndrome in which pathological neovascularization was regulated by VEGF-A-stimulated ADAM17-dependent crosstalk between VEGFR2 and NF-κB [105] and ADAM17 may be involved in a cascade regulating the salivary gland fibrosis observed in Sjögren’s syndrome.

### 2.3. Phosphatidylinositol 3-Kinase (PI3K)/Protein Kinase B (PKB/AKT) Signalling Pathway

The PI3K/AKT signalling pathway regulates cell growth, proliferation, motility, metabolism, and survival [106]. PI3K is a group of lipid kinases associated with the plasma membrane [107], while AKT is a serine/threonine protein kinase activated in response to upstream PI3K [108]. Recently, the PI3K/AKT signalling pathway was implicated as a master regulator for idiopathic pulmonary fibrosis, a disease with an autoimmune aetiology, characterized by a chronic progressive interstitial fibrosis [109,110]. It seems that the overexpression of alpha-smooth muscle actin (α-SMA) in lung fibrosis was related to the activation of PI3K/AKT12, and the interaction between TGF-β and PI3K/AKT promoted lung fibrosis [111]. The activation of PI3K/AKT can determine pulmonary fibrosis by its downstream regulators of mammalian metabolism such as the target of rapamycin (mTOR), hypoxia inducible factor-1α (HIF-1α), and FOX family proteins [108,112]. Recently, the PI3K–AKT pathway was demonstrated to control the release of profibrotic mediators and to disturb the balance between profibrotic and anti-fibrotic mediators [113]. This response was accompanied by abnormal EMT activation, fibroblast proliferation, and fibroblast to myofibroblast transformation [114]. Additionally, myofibroblasts secrete ECM, mainly collagen, which leads to chaotic lung remodelling, and ultimately, progressive pulmonary fibrosis and loss of function. In this context, a pivotal role was exerted by epithelial cells; it is common to observe abnormal epithelial cells, such as bronchial epithelial cells and proliferative type II alveolar epithelial cells, in fibrotic areas in IPF lung biopsies, and studies have shown that alveolar epithelial cell damage is sufficient to cause pulmonary fibrosis [115]. Alveolar epithelial cell apoptosis is frequent in regions with high myofibroblast activity and fibrosis; in addition, these cells produce CTGF, PDGF, and TGF-β which are key fibrogenic mediators [116,117]. The PI3K/AKT signalling pathway has also been reported to be related to liver fibrosis in the course of autoimmune hepatitis, a chronic inflammatory disorder of the liver, characterized by elevation of serum immunoglobulin G (IgG), the presence of autoantibodies, and interface hepatitis on liver histology [118]. Recently, researchers found that PDGF-dependent overexpression of Sparc/osteonectin and kazal-like domain proteoglycan 1 (SPOCK1) promoted hepatic stellate cell activation and proliferation by activating the PI3K/Akt signalling pathway [119]. PI3K/AKT could also regulate angiogenesis by modulating the expression of angiogenic factors such as nitric oxide and angiopoietins and increasing VEGF/VEGFR signalling [120] and enhanced VEGFA/VEGFR2 signalling in liver fibrosis and angiogenesis [121]. In addition, JAK)/STAT-mediated transduction depends on the activation of PI3K/AKT/mTOR signalling [122] and JAK/STAT signals together with TGF-β1/Smad signals promote the EMT process in fibrosis [123].

### 2.4. WNT/β-Catenin

Wingless-related integration site (Wnt) proteins are secreted ligands that signal through the interaction with Frizzled receptors and low-density lipoprotein receptors. Upon binding to their receptors, Wnt proteins induce the stabilization of the transcription factor β-catenin that regulates Wnt target genes transcription [124]. The pro-fibrotic Wnt/β-catenin signalling seems to be active in autoimmune myocarditis [125]. In humans, infections with viruses or parasites, such as Trypanosoma cruzi, often induces heart-specific autoimmune responses, resulting in heart tissue inflammation; this inflammatory condition finally leads to dilated cardiomyopathy, with fibrotic changes in the myocardium and reduced myocardial contractility [126]. Published data have implicated Angiotensin II signalling in the development of autoimmune-mediated myocarditis [127]. Angiotensin II molecular activation was demonstrated to enhance the production of profibrotic TGF-β [128] and Angiotensin II has been implicated in the activation of the pro-fibrotic Wnt that, in turn, activates β-catenin, a transcription factor whose expression is mainly regulated by WNT proteins [125,129,130]. The WNT/β-catenin pathway activates and synergizes with TGF-β1 to mediate myofibroblast activation in autoimmune lung fibrosis [131]. In liver fibrosis, WNT/β-catenin also regulates the expression of vimentin, collagen 1, and fibronectin in hepatic stellate cells induced by TGF-β [132]; however, recently, there seems to be a turnaround derived from the demonstration that constitutively active canonical Wnt/β-catenin signalling confers tolerogenicity to hepatic dendritic cells under steady-state conditions and, therefore, deficiency of canonical Wnt/β-catenin signalling in these cells seems to be responsible for the triggering of autoimmune hepatitis [133].

### 2.5. Peroxisome Proliferator-Activated Receptors (PPARs) Signalling Pathway

Peroxisome proliferator-activated receptors (PPARs) are ligand-dependent transcription factors of the nuclear hormone receptor family [134] which regulate the activation of targeted genes related to lipid and glucose metabolism [135]. There are three PPARs: PPARα, PPARγ, and PPARβ/δ [136]. PPARα is predominantly expressed in brown adipose tissue and the liver [137]. PPAR-α and PPAR-γ activators are involved in the evolution of fibrotic diseases, particularly in cardiac fibrosis [138], renal fibrosis [139], and pulmonary fibrosis [140]. Systemic sclerosis is still a serious disease which is characterized by microvascular dysfunction, autoimmune reactivity, and organ fibrosis [141]. Fibrosis in multiple organs is a final common occurrence in systemic sclerosis [142]. The underlying mechanism of the uncontrolled fibrosis progression in systemic sclerosis remains unclear. However, the altered PPAR-γ expression or function in systemic sclerosis may partly explain the fibrotic program activation in this disease [143]. Several years ago, researchers demonstrated the expression of PPARγ in normal dermal fibroblasts and found that PPARγ inhibition could abrogate the TGFβ-induced collagen gene expression, inhibit myofibroblast differentiation, and repress Smad-dependent gene transcription; in addition, a reduction in PPAR-γ expression was observed in systemic sclerosis [143]. Kohno et al. confirmed the involvement of PPAR with fibrotic pathways because they found correlations between PPARγ ligand expression, the reduction of dermal sclerosis, and the decreased expression levels of CTGF and TGFβ in experimentally induced systemic sclerosis [144]. A further confirmation was derived from the studies conducted by Wu et al. that demonstrated that synthetic PPARγ ligand administration could attenuate inflammation and dermal fibrosis in an experimental animal model of scleroderma [145]. Mice with a knockout of PPARγ in fibroblasts became more susceptible to developing skin fibrosis, as indicated by increased collagen deposition and enhanced inflammation and susceptibility of fibroblasts to pro-fibrotic TGF-β1 signalling [146]. All of these studies established the role of PPARγ in regulating TGF-β-dependent fibrogenesis. The correlation of PPARs with liver fibrosis has been well established. Treatment with PPARα ligands attenuated liver fibrosis in autoimmune hepatitis [147]. The release of TGF-β1 and other inflammatory cytokines are mainly modulated by NF-κB in the development of liver fibrosis. The activation of PPARγ in hepatic stellate cells could block NF-κB by inhibiting the translocation of NF-κB to the nucleus, reducing NF-κB-dependent fibrosis [148].

## 3. Conclusions

Autoimmune diseases are a significant clinical problem because of their chronic nature, which compromises the quality of life, their prevalence in young populations, and the associated healthcare costs. Current therapies, such as the use of specific cytokine antagonists, have shown great promise; however, most of the current therapeutic agents target the terminal phase of the chronic inflammation of these diseases and do not address the fundamental problems that are responsible for the initiation and progression of the autoimmune process which often leads to organ fibrosis. Although controlling the inflammatory state is the most straightforward way to prevent tissue fibrosis, this can be a challenge, as the precise triggers of the inflammation remain unclear. Currently, no effective anti-fibrotic drug is yet available for clinical use in autoimmune patients. In conclusion, the present research topic has collected the most recent findings regarding novel and interconnected molecular pathways of multifaceted chronic inflammatory and fibrotic autoimmune diseases. The increasing elucidation of the molecular and cellular bases for chronic inflammation-associated organ fibrosis could paving the way towards multi-target therapeutic strategies as well as to novel treatments adaptable to various organs sharing similar fibrotic pathogenetic pathways.

## Figures and Tables

**Figure 1 ijms-24-09060-f001:**
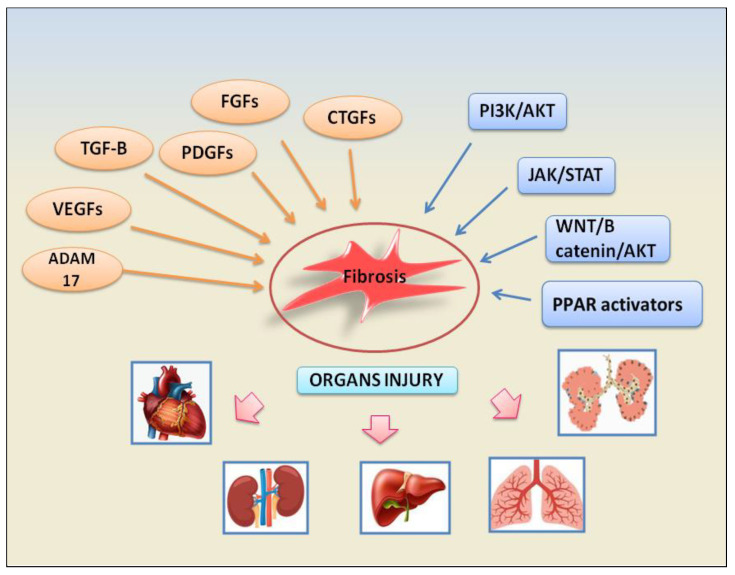
A schematic view of the main factors and signalling pathways involved in organs failure- dependent fibrosis.

**Figure 2 ijms-24-09060-f002:**
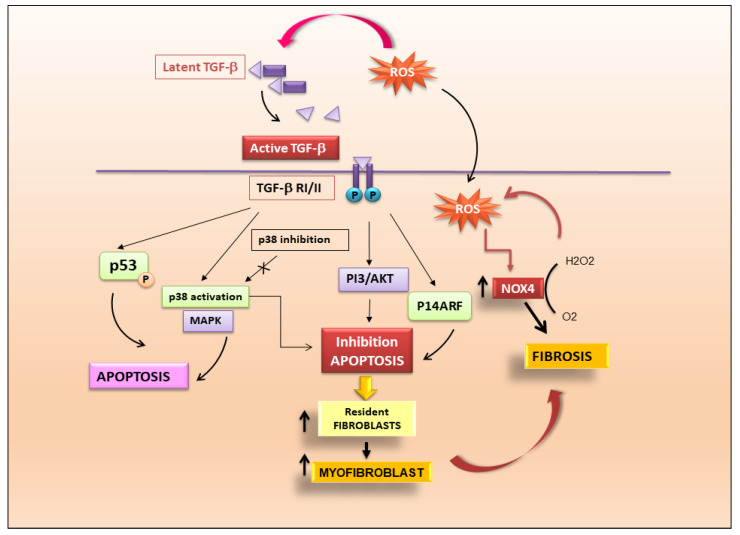
TGF-β is the key modulator in the pathogenesis of fibrosis. Activation of latent TGF-β triggers fibrotic cascades that influence the differentiation of resident fibroblasts to myofibroblasts.

**Figure 3 ijms-24-09060-f003:**
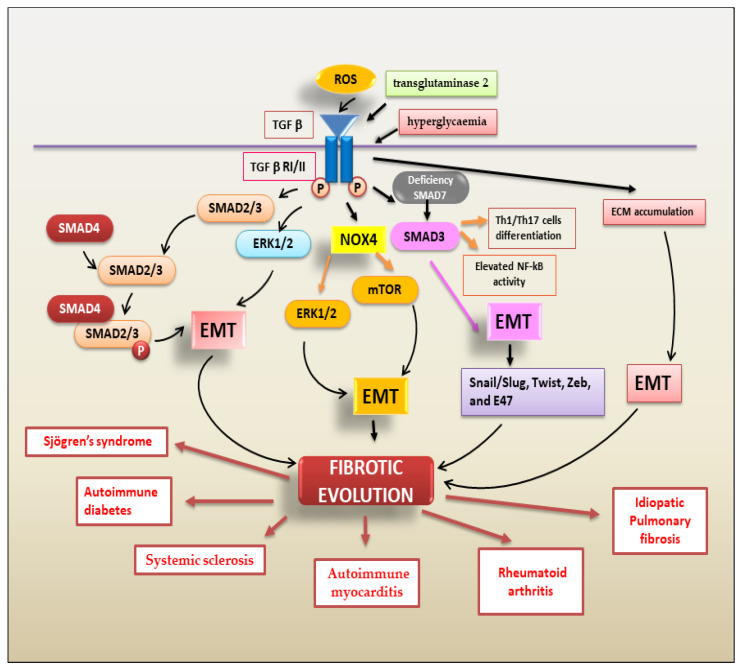
A schematic overview of TGF-β signalling pathways triggering the EMT process that contributes to fibrotic evolution in several autoimmune diseases.

## Data Availability

Not applicable.

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
