# Peer review of "Towards a Unified Approach in Autoimmune Fibrotic Signalling Pathways"

_ijms, 2023, doi:10.3390/ijms24109060_

Round 1
Reviewer 1 Report
Reviewer
Initial comments
This review is very important with a great contribution to understanding the cellular and molecular mechanisms in the fibrotic evolution of several autoimmune diseases.
We know that autoimmune diseases are chronic diseases and are being diagnosed very frequently and are increasing.
The associations of autoimmune diseases are worrying, as many are degenerative diseases that can greatly worsen the quality of life of patients, including some that can lead to death.
Thus this review needs to be published.
Abbreviations – The work is very extensive and has many acronyms, and a list of abbreviations would be advisable for a better understanding of the text.
Title
Towards a unified approach in the autoimmune fibrotic signalling pathways
Comment:
It is suitable
Abstract:
Comment:
It is suitable
1. Introduction
Comment:
It is suitable
2. Molecular mechanisms involved in the fibrotic evolution during autoimmune diseases.
Comment:
It is suitable
2.1. Growth factors and associated signalling pathways
Comment:
It is suitable
2.1.1. TGF-β signalling pathway
Comment:
It is suitable
2.1.1.1. TGF-β-mediated fibrosis through activation of resident fibroblasts
Comment:
It is suitable
2.1.1.2. TGF-β determines fibrosis through the induction of apoptosis
Comment:
It is suitable
2.1.1.3. TGF-β regulates fibrosis through the activation of EMT
Comment:
It is suitable
2.1.2. PDGFs/PDGFRs
Comment:
It is suitable
2.1.3. FGFs/FGFRs
Comment:
It is suitable
2.1.4. VEGFs/VEGFRs
Comment:
It is suitable
2.1.5. CTGF signalling pathway
Comment:
It is suitable
2.2. ADAM17 activation and fibrosis in autoimmune diseases
Comment:
It is suitable
2.3. Phosphatidylinositol 3-kinase (PI3K)/protein kinase B (PKB/AKT) signalling pathway
Comment:
It is suitable
2.4. WNT/β-catenin
Comment:
It is suitable
2.5. Peroxisome proliferator-activated receptors (PPARs) signalling pathway
Comment:
It is suitable
3. Conclusion
Comment:
It is suitable
References
Comment:
It is suitable
Thank you
Author Response
Manuscript ID: ijms-2361517
Title: Towards a unified approach in the autoimmune fibrotic signalling pathways
Authors: Margherita Sisto, Sabrina Lisi
We would like to express our sincere gratitude to the reviewer for her/his positive comments. We respond below in detail to the reviewer’s comment and we hope that the reviewer will find satisfactory our response.
Abbreviations – The work is very extensive and has many acronyms, and a list of abbreviations would be advisable for a better understanding of the text.
We thank the reviewer for her/his positive comment and careful review. As suggest, we added the list of abbreviations as supplementary material. However, to facilitate the reading of the manuscript, the list of abbreviations is given below.
List of abbreviations
ADAM17 (a disintegrin and metalloprotease 17)
AKT (protein-chinasi B)
α-SMA (alpha-smooth muscle actin)
Bax (Bcl-2 Associated protein X)
BCL-XL (B-cell lymphoma-extra large)
CTGF (connective tissue growth factor)
ECM (extracellular matrix)
EGF (epidermal growth factor)
EMT (epithelial mesenchymal transition)
EndMT (endothelial mesenchymal transition)
ERK (extracellular signal-regulated kinase)
FGFs (fibroblast growth factors)
FOX (forkhead box)
FSP1 (ferroptosis suppressor protein 1)
HIF-1α (hypoxia inducible factor-1α)
IAP (inhibitor of apoptosis)
IBD (inflammatory bowel disease)
IL-1 (interleukin-1)
ILDs (interstitial lung diseases)
IPF (idiopathic pulmonary fibrosis)
JAK (Janus kinase)
LAP (latency-associated peptide)
LTBP (latent TGF-β binding protein)
MAPKs (mitogen-activated protein kinases)
mTOR (mammalian/mechanistic target of rapamycin)
NF-κB (nuclear factor kappa-light-chain-enhancer of activated B cells)
Nox (NADPH oxidase)
PDGFs (platelet-derived growth factors)
PI3K (phosphatidylinositol 3-kinase)
PPARS (peroxisome proliferator-activated receptors)
RA (rheumatoid arthritis)
RGD (arginine-glycine-aspartate)
ROS (reactive oxygen species)
SLE (systemic lupus erythematosus)
Smad (suppressor of mothers against decapentaplegic)
SNAIL (zinc finger protein SNAI)
SPOCK1 (kazal-like domain proteoglycan 1)
STAT (signal transducer and activator of transcription)
SS (SjÓ§gren’s syndrome)
SSc (systemic sclerosis)
TG2 (transglutaminase 2)
TGF-β (tumor growth factor Beta)
TNF-α (tumor necrosis factor)
TWIST (Twist-related protein 1)
VEGF (vascular endothelial growth factor)
WNT (wingless/Integrated)
ZEB (zinc Finger E-Box Binding Homeobox)
ZO-1(zonula occludens-1)
Reviewer 2 Report
Although the topic could be interesting, the presentation is a bit confusing: non-fibrotic diseases are mixed with fibrotic diseases, non-autoimmune diseases are included, signalling pathways have been arbitrarily selected without the reasons being clearly explained.
1. The role vasculopathy and autoimmunity are not addressed, as well as epithelial-mesenchymal transition and endothelial-mesenchymal transition.
2. There are many errors systemic sclerosis is described as an autoimmune disease affecting the central nervous system instead of multiple sclerosis.
3. Multiple sclerosis, rheumatoid arthritis, diabetes, Sjogren syndrom are included although they are not considered as fibrotic diseases.
4Idiopathic pulmonary fibrosis is included although it is not an autoimmune disease, all ILD diseases are included although they are not all autoimmune
5. Myocarditis is not a fibrotic disease
Author Response
Manuscript ID: ijms-2361517
Title: Towards a unified approach in the autoimmune fibrotic signalling pathways
Authors: Margherita Sisto, Sabrina Lisi
We would like to express our sincere gratitude to the reviewer for her/his constructive and positive comments and for the very thoughtful critique of our manuscript and are pleased to say that we tried to address all the concerns raised. All changes to the manuscript are highlighted in the text. We respond below in detail to each of the reviewer’s comments and we hope that the reviewer will find satisfactory our responses to the comments.
1.The role vasculopathy and autoimmunity are not addressed, as well as epithelial-mesenchymal transition and endothelial-mesenchymal transition.
We agree that it would be beneficial to look at the role of the vasculopathy and autoimmunity since that vasculitis consists of a collection of heterogenous autoimmune diseases, often with severe life-threatening manifestations. However, we believe that deepening this aspect in this review would make the reading heavier. We have this important research topic in planning in the future work. As concern the epithelial-mesenchymal transition and endothelial-mesenchymal transition, we have published several papers that describe the role of epithelial mesenchymal transition but also endothelial mesenchymal transition in the autoimmune diseases. We reported, here, some related references:
- Sisto M, Lisi S, Ribatti D. The role of the epithelial-to-mesenchymal transition (EMT) in diseases of the salivary glands. Histochem Cell Biol. 2018 Aug;150(2):133-147.
- Sisto M, Ribatti D, Lisi S. Organ Fibrosis and Autoimmunity: The Role of Inflammation in TGFβ-Dependent EMT. Biomolecules. 2021 Feb 18;11(2):310
- Sisto M, Ribatti D, Lisi S. SMADS-Mediate Molecular Mechanisms in Sjögren's Syndrome. Int J Mol Sci. 2021 Mar 21;22(6):3203.
- Sisto M, Ribatti D, Ingravallo G, Lisi S.The Expression of Follistatin-like 1 Protein Is Associated with the Activation of the EMT Program in Sjögren's Syndrome.J Clin Med. 2022 Sep 13;11(18):5368.
- There are many errors systemic sclerosis is described as an autoimmune disease affecting the central nervous system instead of multiple sclerosis
We thank the reviewer to have noted this error. A mistake was occurred in the formulation of the sentence relative to systemic sclerosis and it was deleted.
- Multiple sclerosis, rheumatoid arthritis, diabetes, SjÓ§gren’s syndrome are included although they are not considered as fibrotic diseases.
Multiple sclerosis, rheumatoid arthritis, diabetes and SjÓ§gren’s syndrome are systemic autoimmune diseases characterized by severe and chronic inflammation. Fibrosis, the excessive and inappropriate deposition of extracellular matrix in various tissues, is commonly found in patients with these diseases, and may contribute to various organs dysfunction. Many steps forward have been made in recent years on the knowledge of the cellular and molecular pathways involved in autoimmune fibrosis, discussing the fundamental links between metabolic perturbations, chronic inflammation, autoimmunity and fibrogenic activation. Data collected from recent literature clearly reported that the continued activation of inflammatory-dependent fibrosis is highly detrimental and represents a common final pathway of numerous autoimmune diseases.
- Idiopathic pulmonary fibrosis is included although it is not an autoimmune disease, all ILD diseases are included although they are not all autoimmune
The role of autoimmunity in the pathogenesis of idiopathic pulmonary fibrosis has been the subject of active discussions and investigations in the last years. Circulating antibodies to self antigens have been reported by several groups. Moreover, recently, it was demonstrated the presence of autoantibodies associated with the presence of interstitial lung disease (ILD) involvement in connective tissue diseases. Therefore, actually, a significant proportion of these patients presents autoimmune characteristics.
- Myocarditis is not a fibrotic disease
The importance of fibrosis in organ pathology and dysfunction appears to be increasingly relevant to a variety of distinct diseases. In particular, a number of different cardiac pathologies seem to be caused by a common fibrotic process. Within the heart, this fibrosis is thought to be partially mediated by TGF-β1, a potent stimulator of collagen-production by cardiac fibroblasts. The noxious insult initiating myocarditis causes damage to cardiomyocytes, stimulating the recruitment of circulating immune cells. If the extent of damage results in a loss of cardiomyocytes, the heart repairs itself through the deposition of ECM and myocardial fibrosis. This process can be exacerbated by continued inflammation due to prolonged exposure to the pathogen or toxic agent, T lymphocyte responses to specific antigens and persistent immune responses due to antibodies against or similar to endogenous heart antigens.
Reviewer 3 Report
The manuscript by Sisto and Lisi describes and discusses unique and common signalling pathways leading to fibrosis in chronic autoimmune diseases. It provides important insights into the various pathomechanisms which may contribute to the development of novel therapeutics.
The manuscript is well-written and the figures clearly presented.
Author Response
Manuscript ID: ijms-2361517
Title: Towards a unified approach in the autoimmune fibrotic signalling pathways
Authors: Margherita Sisto, Sabrina Lisi
The manuscript by Sisto and Lisi describes and discusses unique and common signalling pathways leading to fibrosis in chronic autoimmune diseases. It provides important insights into the various pathomechanisms which may contribute to the development of novel therapeutics. The manuscript is well-written and the figures clearly presented.
We thank the reviewer very much for his/her appreciation of the significance and quality of our work.
Reviewer 4 Report
The authors thoroughly reviewed the current progresses in understanding the signaling pathways associated with autoimmune fibrosis in multiple organs. The review article is well written and organized. There are some concerns that need to be addressed.
1. In figure 1, the authors should add the PPAR pathway, which is described at the end of the review. For example, the authors could add “PPAR activators” with an arrow toward the fibrosis.
2. In figure1, the authors have the JAK/STAT pathway. But this pathway is not mentioned in the content of the article. This pathway should be discussed as a subtopic in the text.
3. In lines 70 to 71, the statement that “the interactions between these signaling pathways in fibrosis are depicted in figure 1” is not appropriate and thus needs to be revised. In fact, the figure does not show the interactions between those pathways.
4. In the text, when the EMT and EndMT are mentioned for the first time, they need to be spelled out. The same thing for others.
The authors need to double check the language errors. For example, in line 198, the “in” should not be there.
Author Response
Manuscript ID: ijms-2361517
Title: Towards a unified approach in the autoimmune fibrotic signalling pathways
Authors: Margherita Sisto, Sabrina Lisi
We would like to express our sincere gratitude to the reviewer for her/his constructive and positive comments and for the very thoughtful critique of our manuscript and are pleased to say that we tried to address all the concerns raised. All changes to the manuscript are highlighted in the text. We respond below in detail to each of the reviewer’s comments.
The authors thoroughly reviewed the current progresses in understanding the signaling pathways associated with autoimmune fibrosis in multiple organs. The review article is well written and organized. There are some concerns that need to be addressed.
- In figure 1, the authors should add the PPAR pathway, which is described at the end of the review. For example, the authors could add “PPAR activators” with an arrow toward the fibrosis.
We added in the Figure 1“PPAR activators” as suggested.
- In figure 1, the authors have the JAK/STAT pathway. But this pathway is not mentioned in the content of the article. This pathway should be discussed as a subtopic in the text.
JAK/STAT pathway was mentioned several times in the text, in particular, in the paragraph:”2.3. Phosphatidylinositol 3-kinase (PI3K)/protein kinase B (PKB/AKT) signalling pathway” is reported JAK/STAT pathway as dependent from the activation of PI3K/AKT/mTOR signalling.
- In lines 70 to 71, the statement that “the interactions between these signaling pathways in fibrosis are depicted in figure 1” is not appropriate and thus needs to be revised. In fact, the figure does not show the interactions between those pathways.
The sentence has been rephrased appropriately.
- In the text, when the EMT and EndMT are mentioned for the first time, they need to be spelled out. The same thing for others.
We corrected.
- The authors need to double check the language errors. For example, in line 198, the “in” should not be there.
We done.
Round 2
Reviewer 2 Report
The paper has been improved after incorporating some comments and clarifications of certain points.
I suggest however to remove the part on synovial fibrosis in RA which to my knowledge has never been demonstrated.
Author Response
I thank the reviewer for appreciating the effort made in answering her/his objections and in clarifying the doubts raised. Regarding the last comment, I would like to clarify that in recent years the synovial fibrosis has been studied and demonstrated linked, for example, to rheumatoid arthritis, as demonstrated by published works of which I cite a few:
José Alcaraz M. New potential therapeutic approaches targeting synovial fibroblasts in rheumatoid arthritis. Biochem Pharmacol. 2021 Dec;194:114815.
Zhang, L.; Xing, R.; Huang, Z.; Ding, L.; Zhang, L.; Li, M.; Li, X.; Wang, P.; Mao, J. Synovial Fibrosis Involvement in Osteoarthritis. Front. Med. (Lausanne). 2021, 8, 684389.
Remst, D. F., Blaney Davidson, E. N. and van der Kraan, P. M. (2015). Unravelling osteoarthritis-related synovial fibrosis: A step closer to solving joint stiffness. Rheumatology (Oxford) 54, 1954-1963.
Reviewer 4 Report
My concerns have been addressed.
Author Response
I thank the reviewer for appreciating our manuscript.